# A Comprehensive Review of Natural Flavonoids with Anti-SARS-CoV-2 Activity

**DOI:** 10.3390/molecules28062735

**Published:** 2023-03-17

**Authors:** Jun-Yu Yang, Yi-Xuan Ma, Yan Liu, Xiang-Jun Peng, Xiang-Zhao Chen

**Affiliations:** 1Key Laboratory of Prevention and Treatment of Cardiovascular and Cerebrovascular Diseases of Ministry of Education, Gannan Medical University, Ganzhou 341000, China; 2College of Pharmacy, Gannan Medical University, Ganzhou 341000, China; 3Jiangxi Province Key Laboratory of Biomaterials and Biofabrication for Tissue Engineering, Gannan Medical University, Ganzhou 341000, China

**Keywords:** COVID-19, SARS-CoV-2, flavonoids, treatment, pharmacological effects, synthesis

## Abstract

The COVID-19 pandemic caused by SARS-CoV-2 has majorly impacted public health and economies worldwide. Although several effective vaccines and drugs are now used to prevent and treat COVID-19, natural products, especially flavonoids, showed great therapeutic potential early in the pandemic and thus attracted particular attention. Quercetin, baicalein, baicalin, EGCG (epigallocatechin gallate), and luteolin are among the most studied flavonoids in this field. Flavonoids can directly or indirectly exert antiviral activities, such as the inhibition of virus invasion and the replication and inhibition of viral proteases. In addition, flavonoids can modulate the levels of interferon and proinflammatory factors. We have reviewed the previously reported relevant literature researching the pharmacological anti-SARS-CoV-2 activity of flavonoids where structures, classifications, synthetic pathways, and pharmacological effects are summarized. There is no doubt that flavonoids have great potential in the treatment of COVID-19. However, most of the current research is still in the theoretical stage. More studies are recommended to evaluate the efficacy and safety of flavonoids against SARS-CoV-2.

## 1. Introduction

COVID-19 is a global disease caused by severe acute respiratory syndrome coronavirus 2 (SARS-CoV-2) with a high morbidity and mortality rate. As of November 2022, COVID-19 has spread to 222 countries with more than 600 million confirmed cases and more than 6.5 million cumulative deaths [1]. The COVID-19 outbreak not only endangers human health and livelihoods, but also has a huge impact on global public health systems and economic development.

SARS-CoV-2 is a positive-stranded RNA virus with a characteristic coronavirus spine protein on its outer surface that is capable of droplet transmission [2,3,4]. The viral structure of SARS-CoV-2 has a high degree of similarity to SARS-CoV and MERS-CoV, and it is similarly a zoonotic virus with a long incubation period and high transmission rate [2,5,6,7]. SARS-CoV-2 is highly mutagenic, and there are currently four mutant strains, Alpha, Beta, Gamma, and Delta, with the newly discovered Omicron mutant strain becoming a major global disease and causing strain in just a few months due to its strong infectious and immune-escaping ability [8,9,10,11]. Some common symptoms include fever, headache, shortness of breath, fatigue, cough, nausea, vomiting, diarrhea, and nasal congestion [12,13,14]. Most infected patients are from mild to moderate and do not require special treatment, but those with underlying conditions, such as heart disease, lung disease, and diabetes, as well as the elderly, are more likely to become seriously ill and have a higher mortality rate [13,15,16,17].

In addition to the vaccine, there are currently three new oral crown drugs approved for marketing worldwide, including Paxlovid (Pfizer, New York, USA) [18], Molnupiravir (Merck, Kenilworth, NJ, USA) [19], and the recently approved marketed remdesivir derivative VV116 (Topalliance Biosciences, Shanghai, China) [4,20]. In the early stage of the pandemic, there was a shortage of therapeutic options for COVID-19. Natural products, especially flavonoids, played a huge role in the early stage of the epidemic, showing good results in the treatment and prevention of COVID-19, which attracted widespread attention and research.

Flavonoids are widely found in many fruits and vegetables and have excellent antidiabetic, activity anti-inflammatory, and antiviral activity [21,22]. Flavonoids are mainly obtained through chemical synthesis, biosynthesis, and extraction from plants, and this article provides an overview of the chemical synthesis of several flavonoids. Flavonoid compounds are extremely potent against SRAS-CoV-2 and can be used to treat COVID-19. For example, EGCG inhibits the binding of the virus to the ACE2 receptor [3]. Some flavonoids, such as baicalein, luteolin, and quercetin, can inhibit the activity of 3C-like protease (3CL^pro^), papain-like protease (PL^pro^), and RNA-dependent RNA polymerase (RdRp) [23,24,25]. In addition, various flavonoids such as kaempferol, hesperidin, and isorhamnetin have also been shown to have inhibitory effects on SARS-CoV-2 [26,27,28]. Flavonoids also modulate the levels of inflammatory factors in the body and improve inflammation in COVID-19 patients.

Regarding the research progress of the anti-SARS-CoV-2 role of flavonoids, we summarized in this review, expecting to provide some reference for the development of anti-SARS-CoV-2 drugs. This article will summarize the recent research progress on the anti-SARS-CoV-2 pharmacological activity of flavonoids. It is hoped that this will provide ideas for the development of flavonoid drugs.

## 2. Methods

This paper summarizes the structural classification and pharmacological effects of flavonoids and provides a detailed review of the chemical synthesis of quercetin, EGCG, luteolin, baicalein, and baicalin as well as their anti-SARS-CoV-2 effects.

The literature search was performed through the Elsevier, Web of Science, and PubMed databases. The keywords included: COVID-19; SARS-CoV-2; flavonoids; treatment; pharmacological effects; and synthesis. Non-English literature was selected for exclusion from this review. Other than that, no other restrictions were used in this paper. This article was also accessed on the WHO Coronavirus (COVID-19) Dashboard (https://COVID19.who.int/table, accessed on 30 November 2022).

## 3. Pathogenesis of SARS-CoV-2

SARS-CoV-2 (Figure 1) is a positive-stranded RNA virus with four main structural proteins: the spike protein (S protein), the membrane protein (M protein), the envelope protein (E protein), and the nucleocapsid coat protein (N protein) in addition to some nonstructural proteins (nsps) [29,30,31]. The spike protein is in the outer layer of the virus and has two subunits, S1 and S2. The receptor-binding domain (RBD) of the S1 subunit binds to the ACE2 receptor of the host cell and then, with the involvement of cellular proteins, such as transmembrane serine protease 2 (TMPRSS2), cathepsin B/L (CatB/L), and flavoprotein, fuses with the cell membrane by the S2 subunit and enters the host cell [32,33,34,35]. Upon entry into the cell, the virus produces the polyproteins, 1a and 1b (pp1a and pp1ab) [36]. This is immediately followed by cleavage by 3CL^pro^, also called main protease (M^pro^), and PL^pro^ into 16 nonstructural proteins (nsps) [37,38]. The RdRp is an essential enzyme that can participate in RNA replication and negative-stranded RNA synthesis. The synthesized negative-stranded RNA can be used as a model for mRNA synthesis. The mRNA is translated into a polypeptide chain, which is modified and processed into four structural proteins [39,40]. The viral RNA is wrapped in the N protein and released from the cell by cytokinesis (Figure 2) [41,42,43].

## 4. Classification, Synthesis and Activity of Flavonoids

### 4.1. Structure and Classification of Flavonoids

Flavonoids are a series of polyphenolic compounds found in various plants that have a benzo-γ-pyrone structure and can be synthesized by the phenylpropane pathway. Flavonoids are mostly found in esterified or glycosylated forms, and they form the basic parent nucleus of the C6-C3-C6 structure through 15 carbons (Figure 3) [44].

Depending on the different substitution patterns of the rings, the position of the B ring, and the degree of oxidation of the C ring, there are six main subtypes, including flavones, flavonols, isoflavones, chalcones, flavanes, and anthocyanins (Table 1) [45].

### 4.2. Chemical Synthesis of Flavonoids

Flavonoids can be extracted from plants through biosynthesis and chemical synthesis. This article focuses on the chemical synthesis of flavonoids, emphasizing the synthetic routes of quercetin, baicalein, baicalin, EGCG, and luteolin.

#### 4.2.1. Quercetin

The synthesis route of quercetin is as follows (Figure 4): 2% H_2_SO_4_/H_2_O is added to rutin 1, reacted at 80–90 °C for 4 h, and then filtered; then, the filter cake is washed with water to neutral, then recrystallized in ethanol to provide quercetin [46].

#### 4.2.2. Baicalein and Baicalin

The synthesis route of baicalein is as follows (Figure 5): Compound **3** is obtained by the reaction of trimethoxyphenol 2 with AcOH and BF_3_-Et_2_O at 60 °C for 3 h, which is further condensed with benzaldehyde for 70 h to provide compound **4**. Next, compound **4** is subjected to intramolecular cyclization in the presence of I_2_/DMSO at 100 °C for 2.5 h to provide compound **5**; then, the methyl group is removed using pyridine hydrochloride at 190 °C for 6.5 h to provide baicalein [47].

The synthesis route of baicalin is as follows (Figure 6): Using baicalin as the starting material, the acetylation reaction is carried out in the presence of Ac_2_O and AcONa at 80 °C to obtain acetylated baicalin 6. The acetylated baicalin is refluxed with BnBr in acetone by heating in the presence of KI and K_2_CO_3_ to yield the benzyl-substituted compound **7**. The obtained compound **7** is removed from the benzyl group in THF by the action of Pd(OH)_2_ with H_2_ to provide compound **8**. Additionally, compound **8** is glycosylated with brominated D-glucose in the presence of Ag_2_O at room temperature to provide glycosylated product 9. The removal of the protecting group TBDPS of compound **9** with AcOH and TBAF for 4 h provides compound **10**. The subsequent oxidation of the hydroxyl group of compound **10** with TEMPO and BAIB at room temperature provides product 11. Finally, the protecting group is removed using Mg(OMe)_2_ at room temperature for 3 h to provide the target compound baicalin [48].

#### 4.2.3. EGCG

The synthetic route of EGCG is as follows (Figure 7): Compound **12**, containing allyl alcohol, is oxidized by MnO_2_ at room temperature for 12 h to provide compound **13**. In the presence of HBr, the aldehyde group of compound **13** is cyclized with HOCH_2_CH_2_SH to form cyclic S, O-acetal compound **14**. Then, the oxidation of compound **14** with mCPBA at 0 °C for 8 h provides the oxidized S, O-acetal 15. The additional reaction with compound **15** in the presence of H_2_O is carried out with NBS to provide the bromohydrin compound **16** with stereoisomerism. The brominated alcohol of compound **16** is converted to epoxide 17 by CsCO_3_ in a reaction at 0 °C for 5 h. Additionally, the phenol-containing compound **21** is reacted with compound **17** at 45 °C to provide the cis-ring-opening product 18. In the presence of DIC, the alcohol hydroxyl group of compound **18** is esterified using DMAP; subsequently, the esterified product 19 is treated with an excess of TFAA in the environment of Et_3_SiH, and DCM (1:10) is reacted first at −78 °C for 0.5 h and then at room temperature for 27 h to provide the cyclized product 20. Finally, the protecting group of compound **20** is removed using H_2_/Pd(OH)_2_ at 20 atm to obtain the final product EGCG [49].

#### 4.2.4. Luteolin

The synthesis route of luteolin is as follows (Figure 8): rutin 1 is used as the starting material, the protective agent Na_2_S_2_O_4_ is added, and the target product luteolin is refluxed at 100 °C [50].

### 4.3. Pharmacological Activity of Flavonoid Compounds

#### 4.3.1. Anti-Inflammatory Effect

Cytokine storm is a feature of the inflammatory response induced by SARS-CoV-2 [51,52]. Flavonoids can target pathways, such as NF-κB, MAPK, ERK, and Akt, and can also reduce the release of inflammatory cytokines, which play an anti-inflammatory role [53].

A study showed that quercetin inhibited MC903-induced atopic dermatitis and improved arthritis by reducing proinflammatory factors [54,55]. Kaempferol blocked the ROS/NF-κB signaling pathway and reduced the inflammatory response in atherosclerosis [56]. In addition, kaempferol also regulated the expression of adipogenesis and reduced lipid accumulation in CCAAT/enhancer binding protein α (CEBPA) by upregulating the mRNA expression of Pnpla2 and Lipe [57]. Janumetin prevented neuroinflammation caused by sleep deprivation and also treated atopic dermatitis caused by obesity [58,59]. Both chrysin and lignans have been shown to have preventive effects against ochratoxin A-induced gastrointestinal inflammation in vitro [60]. Oroxylin A has a therapeutic effect on collagen-induced arthritis (CIA). In a mouse model of CIA, researchers found that mice treated with Oroxylin A had significantly lower levels of inflammatory factors in their serum and reduced somatic damage caused by arthritis [61]. Rutin is reportedly effective in treating colitis. Animal research discovered that no additional medication was required, and only 0.1% rutin was added to the daily diet and fed for 2 weeks. The concentration of proinflammatory factors in the serum of the mice was lowered, and the symptoms of colitis were reduced [62]. Saeedi-Boroujeni’s study reported that quercetin was able to affect the thioredoxin-interacting protein (TXNIP), thereby inhibiting NLRP3 inflammatory vesicles and achieving an inflammatory suppressive effect [63]. Clinical studies have shown that adjuvant therapy with quercetin in early COVID-19 patients significantly reduced the release of proinflammatory factors, such as TNF-β and IL-1β, and alleviated inflammation in patients with COVID-19 [43].

#### 4.3.2. Antiviral Effect

Flavonoids are effective throughout the life of a virus. They can act through the inhibition of the virus cell entrance, replication of the viral gene set, translation and processing of proteins, and release of the virus from the cell [64].

It was demonstrated that both biochanin A and baicalin were able to inhibit H5N1 virus replication in A549 cells, but the mechanisms of action were different [65,66]. Both fisetin and rutin blocked viral replication by inhibiting the 3CL^pro^ activity of enterovirus A71 [67]. Formononetin modulated COX-2/PGE₂ expression, thereby inhibiting the replication with enterovirus A71 [68]. Silymarin reportedly has anti-dengue virus activity in vitro and hepatoprotective properties for HCV infection treatment [69,70]. An in vitro study has shown that flavonoids isolated from the above-ground parts from *Marcetia taxifolia* are effective against herpes simplex virus, poliovirus, and hepatitis B virus [71]. The total flavonoids extracted from Robinia pseudoacacia cv. idaho showed the significant inhibition of herpes simplex virus type 1 and enterovirus type 71 with therapeutic indices of 113.8 and 46.2 [72]. It was shown that selenium functionalization with quercetin enhanced the inhibitory effect on M^pro^ and that quercetin significantly inhibited SARS-CoV-2 infection at higher concentrations, while quercetin derivatives inhibited the viral infection at low concentrations [23]. Baicalein and baicalin were able to bind to SARS-CoV-2 RdRp, causing the RdRp to be unable to participate in the RNA replication process of the virus [73]. Several studies in vitro have shown that EGCG can prevent the RBD region of SARS-CoV-2 from binding to the ACE2 receptor, thus preventing the virus from entering cells [74,75]. Numerous studies have proved that kaempferol, catechin, rutin, hesperidin, naringenin, and luteolin all have anti-coronaviral effects.

#### 4.3.3. Antidiabetic Effect

Diabetic patients infected by SARS-CoV-2 will lead to exacerbation of the disease. Reducing the effect of diabetes on COVID-19 is particularly important. Flavonoids are extremely useful in treating diabetes and diabetic complications. They can exert their therapeutic effects on diabetes by enhancing insulin secretion, regulating glucose metabolism, and reducing inflammation and oxidative stress [76,77,78].

A study showed that EGCG and quercetin reduced insulin resistance, both in vivo and in vitro, and also reduced glucose metabolism in the liver [79]. A study in a rat model of diabetes showed that EGCG improved diabetes in rats and improved streptozotocin-induced complications [80]. Catechin is a naturally occurring product that has also been shown to have anti-diabetic activity. Treating rats with Eudragit particles loaded with catechins significantly reduced the concentration of blood glucose [81]. Clinical studies have found that vitamin C and rutin, when administered, significantly lowered fasting blood sugar in those with type 2 diabetes. In addition, rutin can improve neuropathy in diabetic patients [82,83]. A study exhibited that flavonoids extracted from Cistus laurifolius L. inhibited both α-glucosidase and α-amylase in vivo and vitro [84]. In a mouse model of streptozotocin-induced diabetes, the combination of Astragalus polysaccharides and Astragalus flavonoids significantly improved the function of insulin and thus exerted an anti-diabetic effect [85].

The collected evidence suggests that flavonoids can exert their antidiabetic effects through different mechanisms. However, there are no published studies about the treatment of COVID-19 diabetic patients with flavonoids.

#### 4.3.4. Anticancer Effect

Studies showed that cancer patients with SARS-CoV-2 had a higher mortality rate [86]. Flavonoids can also exert antitumor effects through mechanisms such as antioxidants, COX-2 inhibition, immunomodulation, affecting cell cycle effects, apoptosis induction in cancer cells, tumor angiogenesis prevention, and telomerase activity inhibition [87,88].

In a study of the agent-sensitive LoVo cell lines and their agent-resistant LoVo/Dx subline model, baicalein and luteolin inhibited the development of colon cancer cells [89]. Baicalein also improves the effectiveness of cisplatin in human lung cancer cells [90]. Apigenin had been shown to inhibit the PI3K/Akt/mTOR pathway during viral accretion, thereby suppressing viral accretion [91]. Oroxylin A was shown to have a beneficial therapeutic effect on breast cancer by specifically binding to α-actinins 1 (ACTN1), thereby inhibiting ACTN1 expression to prevent cancer cell metastasis [92]. Oroxylin A also showed a positive inhibitory effect on lung cancer by suppressing lung cancer cell proliferation and metastasis in vivo [93]. Latifolin blocked cell growth, division, migration, invasion, and adhesion in oral squamous cell carcinoma by targeting PI10K/AKT/mTOR/p3S70K signaling [94]. EGCG and BAY11-7082 synergistically acted for the suppression of lung cancer cell proliferation both in vitro and in vivo [95].

Although SRAS-CoV-2 does not directly cause cancer, cancer patients infected with SRAS-CoV-2 have more severe symptoms and a higher lethality rate. Further investigating cancer cell inhibition would help improve COVID-19 patients’ symptoms.

## 5. Anti-SARS-CoV-2 Activity of Flavonoids

### 5.1. Anti-SARS-CoV-2 Pharmacological Effects of Quercetin

Quercetin is a flavanol compound that is widely found in various plants, mainly in the form of glycosides [96]. Studies have shown that quercetin has strong anti-inflammatory, antiviral, and immunomodulatory activities [97,98,99]. Cytokine storm is a feature of the inflammatory response induced by SARS-CoV-2 and is a major cause of death by COVID-19 [51,52]. According to relevant studies, NLRP3 inflammatory vesicles play an important role in inflammation [100]. Therefore, inhibiting NLRP3 inflammatory vesicle activation can effectively suppress the inflammatory response. Saeedi-Boroujeni’s study reported that quercetin could affect the TXNIP, thereby inhibiting NLRP3 inflammatory vesicles and achieving an inflammatory suppressive effect [63]. Clinical studies have demonstrated that the adjuvant treatment of early COVID-19 with quercetin significantly reduces the viral load and release of proinflammatory factors. Meanwhile, quercetin combined with antivirals for COVID-19 reduces mortality and the length of stay [101,102]. Meanwhile, Mangiavacchi et al. have shown that selenium functionalization with quercetin enhances the inhibitory effect on M^pro^. According to the RT-qPCR results, quercetin at higher concentrations significantly inhibited SARS-CoV-2 infection, while quercetin derivatives inhibited the viral infection at low concentrations [23].

### 5.2. Anti-SARS-CoV-2 Pharmacological Effects of Baicalein and Baicalin

Baicalein and baicalin are both flavonoid compounds extracted from the dried roots from scutellaria baicalensis, which have various pharmacological effects, including anti-inflammatory, antiviral, antibacterial, hepatoprotective, and choleretic [103,104]. Liu et al. investigated the antiviral activity of baicalein against SARS-CoV-2 using RT-qPCR and showed that baicalein was able to inhibit the replication of SARS-CoV-2 in Verb cells in vitro with an EC50 of 2.9 μM, SI > 172 (SI = CC50/EC50) [105]. Su et al. screened the novel inhibitors of 3CL^pro^ using a FRET protease assay and discovered that baicalein and baicalin exhibit a significant inhibitory effect on 3CL^pro^ [24]. In addition, Keivan Zandi et al. found that baicalein and baicalin are able to bind to SARS-CoV-2 RdRp, causing RdRp to be unable to participate in the virus^’^ RNA replication process; SARS-CoV-2 RdRp inhibition was first demonstrated in a study by Keivan Zandi et al. [73].

### 5.3. Anti-SARS-CoV-2 Pharmacological Action of EGCG

EGCG is a flavonoid extracted from green tea with various pharmacological activities, such as antibacterial, antiviral, antioxidant, and anti-inflammatory effects [106]. SARS-CoV-2 can enter host cells by binding to the ACE2 receptor via the surface S protein. Several studies in vitro have shown that EGCG can prevent the RBD region of SARS-CoV-2 from binding to the ACE2 receptor, thus preventing the virus from entering cells with a low cytotoxicity [3,74,75,107,108]. EGCG can inhibit the replication of SARS-CoV-2 by inhibiting certain key enzymes in the RNA replication process. For example, Nsp15 (U-specific endoribonuclease) can cleave the polyU sequence in viral RNA, thus interfering with the host’s immune system and enabling the virus to undergo immune escape. Additionally, Nsp15 is significant in virus replication [109,110]. An in vitro study by Hong et al. exhibited that EGCG significantly inhibited the Nsp15 activity of SARS-CoV-2, with drug concentrations below 1 μg/mL completely inhibiting the activity of Nsp15 and thereby inhibiting virus replication in cells [111]. Furthermore, some studies showed that EGCG inhibits M^pro^ in vitro, thereby inhibiting virus replication [112,113,114].

### 5.4. Anti-SARS-CoV-2 Pharmacological Action of Luteolin

Luteolin, mostly in the form of glycosides, exists in a variety of plants, has anti-inflammatory, antiallergic, antiviral, antitumor, antibacterial, and other pharmacological activities, and is often clinically used for its anti-inflammatory effects, coughs, and expectorants [115,116,117]. In vitro studies have shown that as little as 20 μM luteolin has an inhibitory effect on 3CL^pro^. Additionally, luteolin also inhibits RdRp activity [25]. Xiao et al., in a SARS-CoV-2 pseudovirus experiment, discovered that luteolin is able to bind to the S protein and significantly inhibits the entry of SARS-CoV-2 into cells with an EC50 less than 7 μmol/L [118]. COVID-19 can cause a loss of smell or taste. A clinical study by L. D’Ascanio et al. showed that daily oral supplementation of palmitoylethanolamide and luteolin was able to restore the patient’s sense of smell [119]. In addition, clinical studies by Lisa O’Byrne et al. also exhibited that palmitoylethanolamide and luteolin supplementation could intervene to treat olfactory dysfunction in those suffering from COVID-19 [120].

### 5.5. Other Flavonoids with Anti-SARS-CoV-2 Activity

Kaempferol can inhibit the activity of 3CL^pro^. The study by Abbas Khan and Wang Heng et al. exhibited that a 62.5–125 μg/mL concentration of kaempferol can significantly shorten the cytopathic effects (CPE) caused by Vero E6 cell infection in vitro [27].

Both hesperidin and hesperitin can inhibit the activity of TMPRSS2 and ACE2 by binding to them. Additionally, they can block the SARA-CoV-2 S protein from binding the ACE2 receptor and prevent SARA-CoV-2 from entering the cell [28]. Hesperidin also blocks the AKT pathway and inhibits Ang II-induced collagen expression and cardiac fibroblast proliferation during COVID-19 infection [121]. Clinical studies have also shown that hesperidin can reduce some of the clinical symptoms of COVID-19, such as shortness of breath, cough, decreased or even absent taste, and fever [122].

Furthermore, an in vitro study revealed that isorhamnetin also interacts with ACE2 to exert anti-SARS-CoV-2 activity [26]. It was found that naringenin exhibits potent anti-SARS-CoV-2 activity in vitro by inhibiting M^pro^ [123,124].

Diosmin, biochanin A, and silymarin are able to reduce the inflammatory response and alleviate inflammation by decreasing cytokine levels in patients with COVID-19 [125,126,127].

The in vivo and in vitro activity studies of some flavonoid compounds are shown in Table 2 and Table 3 below.

## 6. Conclusions

The COVID-19 outbreak endangers human health and livelihoods and heavily impacts global public health systems and economic development. Although there are vaccines and specific drugs to treat COVID-19, such as Paxlovid, Molnupiravir, and VV116, due to the instability of the virus, which is prone to immune escape, researchers need to investigate more drugs and options for treating COVID-19.

Flavonoids are widely found in various plants and have a significant effect on both the prevention and treatment of SARS-CoV-2. Several in vivo and in vitro studies have demonstrated that flavonoids exhibit excellent antiviral activity against SARS-CoV-2 and can inhibit SARS-CoV-2 by inhibiting key viral targets, including the ACE2 receptor, TMPRSS2, M^pro^, RdRp, S protein RBD, etc. In addition, flavonoids also have an inhibitory effect on inflammation caused by SARS-CoV-2, inhibiting the production and release of various proinflammatory factors in the inflammatory response [151]. At the same time, flavonoids improve some of the clinical symptoms of COVID-19.

Although many studies have reported flavonoids’ anti-SARS-CoV-2 effects, most of them are theoretical studies. Only a few in vivo and clinical studies are available. Thus, more applied experimental studies are needed to explore the drugs’ safety and efficacy. Secondly, the bioavailability of the ingested compounds is limited, and how to improve the bioavailability of the compounds is also an issue to be considered. Choosing the right route of administration and preparing the drug into formulations can improve a drug’s bioavailability [152]. While there are still some problems, flavonoids can undoubtedly show anti-SARS-CoV-2 effects through direct or indirect pathways. Thus, they represent a group of promising anti-SARS-CoV-2 compounds. Flavonoids have excellent medicinal potential. We are expecting more studies to explore the medicinal value of flavonoids and to develop flavonoid drugs.

## Figures and Tables

**Figure 1 molecules-28-02735-f001:**
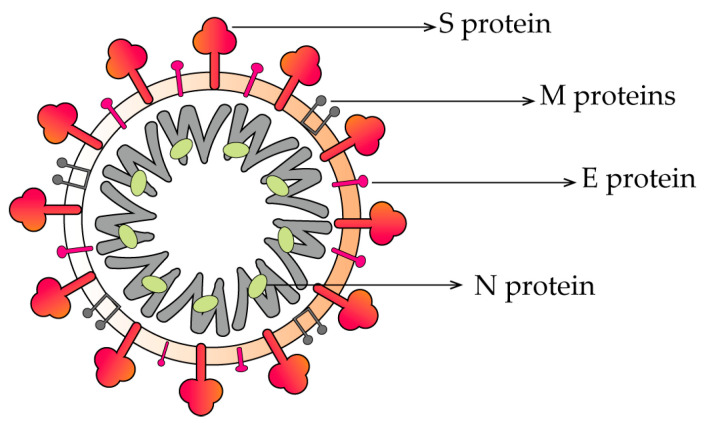
Structure of SARS-CoV-2. Abbreviations: S: Spike; M: Membrane; E: Envelope; N: Nucleocapsid.

**Figure 2 molecules-28-02735-f002:**
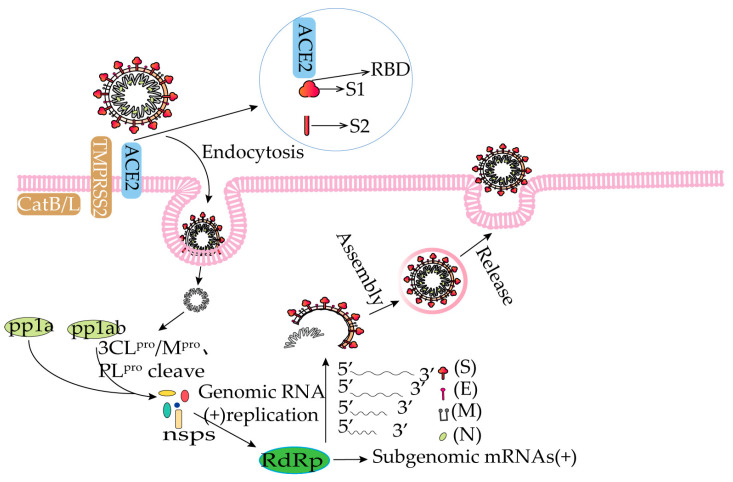
Pathogenesis of SARS-CoV-2. Abbreviations: ACE2: angiotensin converting enzyme 2; TMPRSS2: transmembrane serine protease 2; CatB/L: cathepsin B/L; RBD: receptor-binding domain; nsps: nonstructural proteins; pp1a: polyproteins 1a; pp1ab: polyproteins 1ab; 3CL^pro^: 3C-like protease; M^pro^: main protease; PL^pro^: papain-like protease; RdRp: RNA-dependent RNA polymerase.

**Figure 3 molecules-28-02735-f003:**
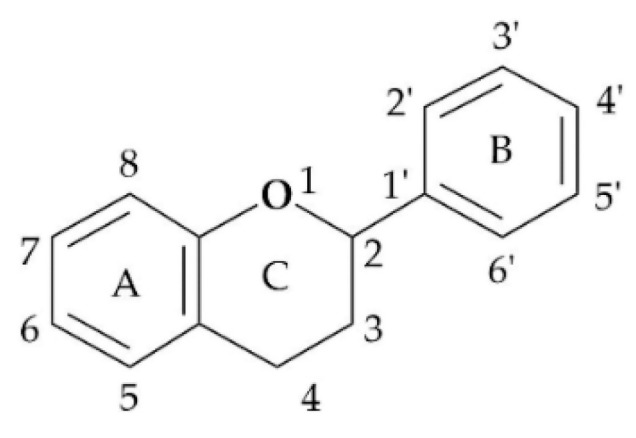
Basic skeleton of flavonoids. (**A**) and (**B**): Benzene ring. (**C**): Closed pyran ring.

**Figure 4 molecules-28-02735-f004:**
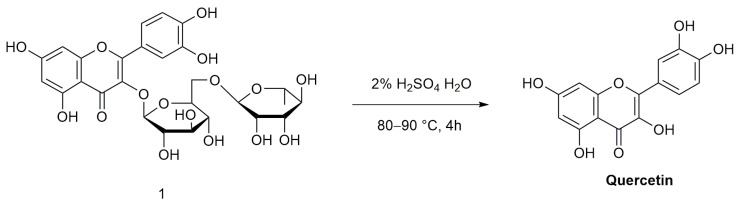
Synthetic route of quercetin.

**Figure 5 molecules-28-02735-f005:**
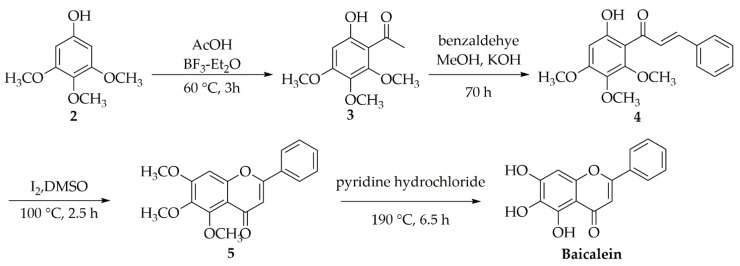
Synthetic route of baicalein.

**Figure 6 molecules-28-02735-f006:**
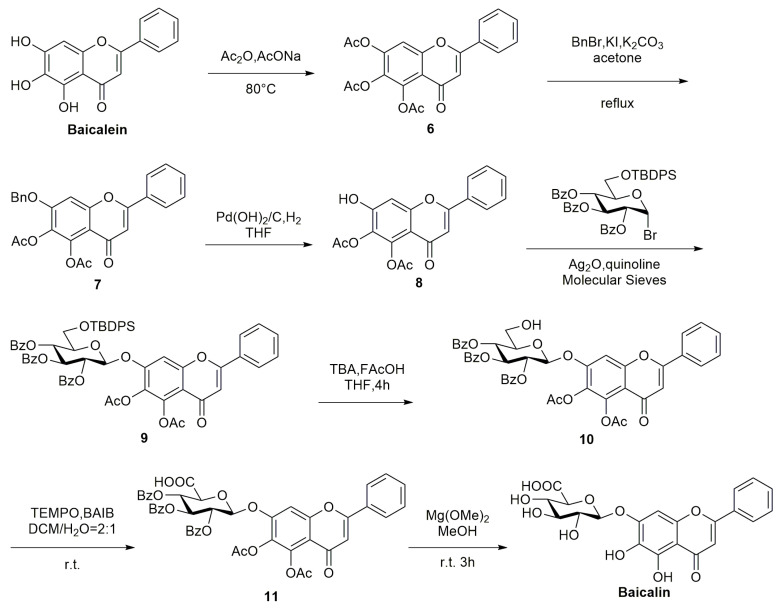
Synthetic route of baicalin.

**Figure 7 molecules-28-02735-f007:**
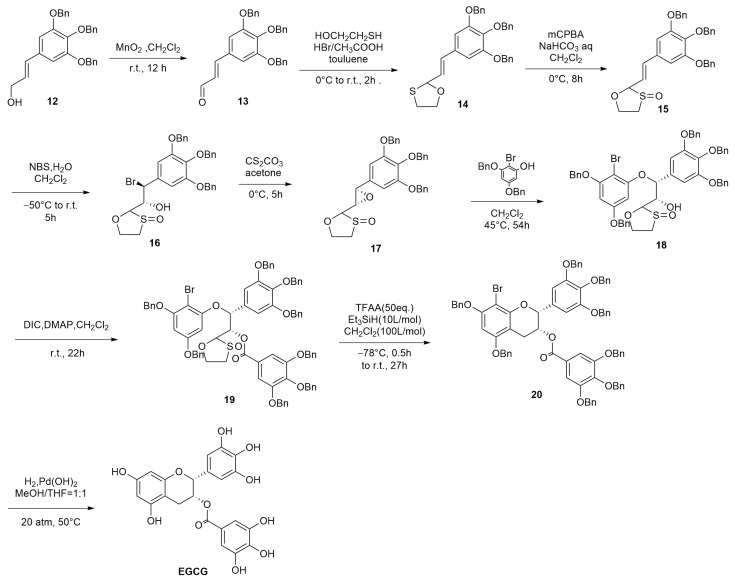
Synthetic route of EGCG.

**Figure 8 molecules-28-02735-f008:**
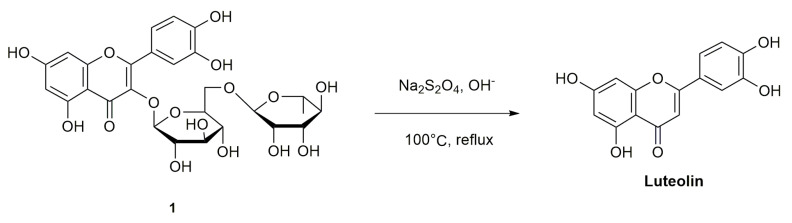
Synthetic route of luteolin.

**Table 1 molecules-28-02735-t001:** Structure of flavonoids.

Flavonoids	Basic Structure	Example
Flavones	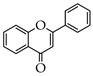	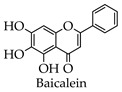	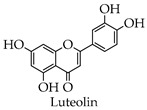	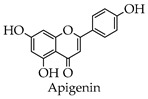
Flavonols	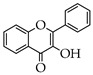	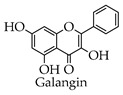	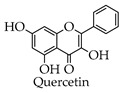	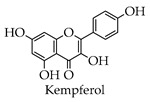
Isoflavones	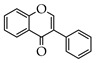	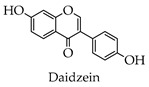	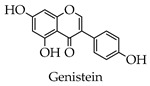	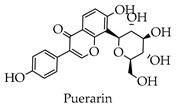
Chalcones	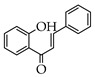	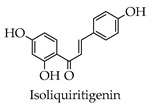	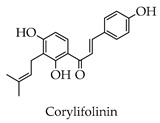	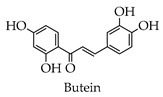
Flavanes	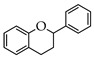	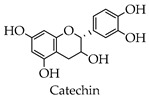	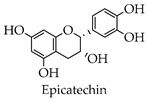	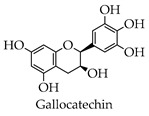
Anthocyanins	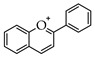	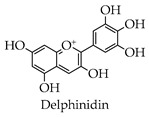	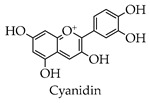	

**Table 2 molecules-28-02735-t002:** Study on the in vitro activity of flavonoids.

Compound	In Vitro Models	Conc.	Effects	Mechanism of Action	References
Baicalin	HepG2 cells	16 μM	Pneumonia	Inhibits NLRP3 inflammatory vesicles	[128]
	Rat IEC-6 cells	10.0 μg/mL	Inflammatory bowel disease	Inhibits IL-6 and TNF-α inflammatory factor levels	[129]
	KOPN-8, RCH-ACV, SEM, RS4-11, NALM-6	10 mg/mL	Acute B-lymphoblastic leukemia	Inhibits the glycogen synthase kinase 3β and induces cell cycle arrest by upregulating p27Kip1	[130]
	MCF-7 and MDA-MB-231 cells	30 μM	Breast cancer	Inhibits breast cancer cell proliferation and induces G1/S arrest in cells	[131]
	Vero, BHK-21, and HEK 293T cells	100 μM	Chikungunya virus	—	[132]
Baicalein and baicalin	Vero CCL-81 cells	20 μM	SARS-CoV-2	Inhibits SARS-CoV-2 RNA-dependent RNA polymerase activity	[105]
Baicalein	Vero E6 cell	>2.0 μM	COVID-19	Inhibits virus replication in vivo and reduces serum levels of IL-1β and TNF-α	[133]
	Vero cell lines	50 μM	SARS-CoV-2	Inhibits SARS-CoV-2 replication and its 3C-like protease	[105]
	HCT116 cells	50 μM	Colorectal cancer	Induces apoptosis in human colon cancer cells	[134]
Silymarin	Vero cells	749.70 µg/mL	Dengue virus	Binds to the surface protein of the virus to prevent entry into the cell	[70]
Luteolin	ARPE-19 cells	50 μM	Age-related macular degeneration	Decreases levels of IL− 8 and IL− 6	[135]
	MDCK cells and Vero cells	From 3.75 to 240 μM	Influenza A virus	Inhibits the expression of β-COP protein	[136]
	Enzymatic inhibition assay	4.6 μM	COVID-19	Inhibits SARS-CoV-2 RNA-dependent RNA polymerase activity	[25]
EGCG	Peripheral blood lymphocytes	50 μM	HIV	—	[137]
	live SARS-CoV-2 strain	1 μg/mL	SARS-CoV-2	Inhibits Nsp15 activity	[111]
	*E. coli*	7.58 μg/ mL	SARS-CoV-2	Inhibits the activity of 3CL^pro^	[112]
Formononetin	Vero cells	14.91 μmol/L	Enterovirus 71	Inhibits EV71-induced COX-2 expression and PGE_2_ production via MAPKs pathway	[68]
Biochanin A	A549 cells	5 μM	H5N1 influenza A viruses	Interferes with AKT, ERK ^1^/_2_, and NFκB activation to inhibit viral replication	[65]
Quercetin	T-REx™-293 cell line	10 μg/mL	HCV	Inhibits NS3 protease, thereby inhibiting viral replication	[138]
Naringenin	Huh-7.5 cells	25 μM	HCV	Reduces production of intracellular viral proteins	[139]

**Table 3 molecules-28-02735-t003:** Study on the in vivo activity of flavonoids.

Compound	In Vivo Models	Conc.	Effects	Mechanism of Action	References
Baicalin	Male C57BL/6 mice	21 mg/kg	Arthritis	Antagonism of Th-17 cells	[140]
	Female BALB/C nude mice	200 mg/kg	Breast cancer	Inhibits breast cancer cell proliferation and induces G1/S arrest in cells	[131]
	Six-week-old male C57BL/6J mice	21 mg/kg	Diabetes	Activates AKT/AS160/GLUT4 pathway	[141]
Baicalein	Male SD rats	200 mg/kg	COVID-19	Inhibits virus replication in vivo and reduces serum levels of IL-1β and TNF-α	[133]
	Male ICR mice	1, 5, and 10 mg/kg	Colorectal cancer	Induction of apoptosis in human colon cancer cells	[134]
	Female C57BL/6 mice	0.8 mg/mouse nine times	Bladder cancer	Reduces expression of cyclin D1 by inhibiting new protein synthesis and promoting proteasomal degradation and reduces expression of cyclin B1 by inhibiting new protein synthesis.	[142]
	Male C57BL/6 mice	0.5 mg/kg	Type 2 diabetes	Improves the viability and insulin secretion of fine beta cells and human pancreatic islets	[143]
Luteolin	Male ICR mice	100 mg/kg	Acute pancreatitis	Suppresses the activation of the NF-κB pathway	[144]
	Male BALB/cN mice	5mg/kg	Nephrotoxicity	Reduces Pt accumulation in the kidney to improve oxidative stress and inflammation	[145]
	BALB/c nude mice	40 mg/kg	Gastric carcinoma	Downregulates VEGF-A and MMP-9 and decreases the immune response	[146]
EGCG	Male C57BL/6 mice	10 mg/kg	SARS-CoV-2	Inhibits virus replication	[114]
	Male ICR mice	100 mg/kg	Diabetic nephropathy	Inhibits increased OPN expression, reduces serum creatinine, and causes proteinuria and normalized morphological changes in STZ-induced diabetic nephropathy	[147]
Oroxylin A	Female BALB/c nude mice	200 mg/kg	Lung cancer	Inhibits the activation of ERK signaling and inhibits snail protein content and EMT	[93]
	Male DBA/1 mice	10 mg/kg	Rheumatoid arthritis	Reduces serum levels of anti-CII Abs, IL-1β, IL-6, TNFα, and IL-17	[61]
Tiliroside	Male C57BL/6 mice	5 mg/kg, 15 mg/kg, 300 mg/kg Dose-dependent manner	Type 2 diabetes	Regulates miR-27 expression and inhibits glycoisomerization	[148]
Biochanin A	Male C57BL/6 mice	12.5, 25, and 50 mg/kg	Acute lung injury	Inhibits TLR4/NF-κB signaling pathway activation and increases PPAR-γ expression in the lungs	[126]
Isorhamnetin	Male ICR mice	1 nM	Diabetes	Activates JAK/STAT pathway to increase glucose uptake by muscle cells	[149]
Quercetin	Male ICR mice	0.1 nM and 1 nM	Diabetes	Activates CaMKKβ/AMPK signaling pathway and activates IRS3/PI10K/Akt signaling	[149]
Naringenin	Pregnant Sprague–Dawley rats	25, 50, or 100 mg/kg	Neuroapoptosis	Regulates PI3/Akt/PTEN signaling pathway and inhibits NF-κB-mediated inflammation	[150]

## Data Availability

Not applicable.

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
