# Peer review of "A Comprehensive Review of Natural Flavonoids with Anti-SARS-CoV-2 Activity"

_molecules, 2023, doi:10.3390/molecules28062735_

Round 1

Reviewer 1 Report

This review addresses the latest research advances in the pharmacological activity of flavonoids in anti-SARS-CoV-2, focusing on the anti-SARS-CoV-2 action of several flavonoids and their chemical synthesis, which may provide some ideas for the development of safe and effective therapeutic agents on COVID-19.

Natural products, especially flavonoids, played a huge role in the early phase of the epidemic, showing good results in the treatment and prevention of COVID-19. The article reviews the progress of research on flavonoid compounds in anti-SARS-CoV-2 in recent years and focuses on the pharmacological activity and synthesis of quercetin, baicalein and baicalin, EGCG (epigallocatechin gallate), luteolin.

The review carried out by the authors shows that most of the studies describing the effect of flavonoids on SARS-CoV-2 are theoretical studies and few in vivo studies and clinical trials are available. Therefore, further empirical studies are needed to investigate the safety and efficacy of the drugs. Secondly, the bioavailability of ingested compounds is limited and the question of how to improve the bioavailability of compounds is also an issue to be considered.

This literature review provides important information on natural flavonoids with potential anti-SARS-CoV-2 properties and although there are still problems to overcome, it should be emphasized that they can play a role as compounds acting directly against SARS-CoV-2 and/or as compounds supportive drugs marketed worldwide, including Paxlovid (Pfizer), Molnupiravir (Merck Sharp & Dohme) and the recently approved raltegravir derivative VV116 (Juniper Biologicals).

The work is interesting and worth publishing in Molecules.

Author Response

Thanks to the Reviewer's summary of our work.

  •  

Reviewer 2 Report

I recommend making the following adjustments:
1. The abstract should be expanded as it is too uninformative. To clearly indicate what has been done so far, what is the purpose of the present study and what are the future perspectives on the mentioned topic.
2. The introduction is too short. To indicate current studies on the isolation and data on the pharmacological effect of the flavonoids indicated by the authors.
3. There is no logical connection between chapters 3 and 4. To make such an idea, consider combining them under the example title Structure, classification and action of flavonoids on SARS-CoV pathogens 2. To give more information about the proposed classification - factors and/or terms of the basis on which it is made.
4. I do not see the connection of chapters 4.1 and 4.2 pathogens of the SARS-Covid 2 type.
4. I recommend expanding parts 4.3, 4.4 and 4.5, indicating if possible studies done on SARS-COVID pathogens.
5. Chapter 5 on isolation of flavonoids should be moved before chapter 4.

Reviewer 3 Report

Uniform same font in overall MS

Professional English editing is needed to correct some errors .

Whole MS is in plagiarism

Reduce the % of plagiarism upto 10%

Make table for in-vivo and in-vitro flavonoids compounds

Table for pharmacological activity of flavonoids

Line 75-79: Rewrite

Line 87-90: Rewrite

Line 172-180- Rewrite

Line 210: Correct reference

The writing could be improved by strengthening the connectivity between paragraphs. There are several places where new topics are introduced and connections to the previous subject are not clear. Read whole manuscript and correct wherever required.

Introduction:

The introduction does not clearly state the purpose of the research – please amend.

Conclusions

The conclusions are too general, format according to future aspects. Please make them more specific.

Carefully read whole manuscript line by line and improve the sentence formation

Cross check all references and style of reference according to Journal format, use abbreviation of journal name in reference

Reviewer 4 Report

Manuscript “A Comprehensive Review of Natural Flavonoids with Anti-SARS-CoV-2” is a good and informative study but had some minor corrections. Below are some comments/suggestion for the authors to improve its quality:

  1. Language should be improvised.
  2. In Abstract section methodology of study should be included.
  3. Also abstract section should include some conclusive statements.
  4. Introduction section must include the need and significance of study.
  5. Introduction section must include some information on the importance of flavonoids to be taken for treatment.
  6. Main manuscript also include methodology section; clearly stating inclusion and exclusion criteria.
  7. There are more studies available that can be cited which includes some pathways details of compounds present in Section 4. for eg  Doi: https://doi.org/10.1016/j.neubiorev.2022.104795, etc. Some works suggesting mechanistic roles of Flavonoids in infection prevention should be cited: 10.1016/j.tifs.2020.10.020, 10.3390/biom9050161.
  8. State the temperature and environmental conditions used in all the reactions mentioned, to synthesize  different flavonoids.
  9. Also in reactions of synthetic route at each step new compounds are formed, if possible state the name of that compounds instead of numbers.
  10. Similar studies are present, please state the novelty of your study. for eg Doi: 10.1007/s11101-021-09759-z
  11. The reason of selecting these specific flavonoids in the study. As other flavonoids information are also available. For eg https://doi.org/10.3390/molecules27196374

12.   Few recent relevant studies are present that have to be cited. For eg: Doi: http://dx.doi.org/10.2174/1573406418666220829144746, https://doi.org/10.1155/2022/6003869, 10.3389/fcimb.2022.929430,  10.1016/j.ijsu.2022.106818, 10.1093/jtm/taad015, 10.1002/jmv.27626, 10.3389/fphar.2021.575877.

Round 2

Reviewer 2 Report

The new version is better than the first one . i accept it in the present form.

Reviewer 3 Report

Accepted in Present form